# Effect of MWCNT Surface Functionalisation and Distribution on Compressive Properties of Kenaf and Hybrid Kenaf/Glass Fibres Reinforced Polymer Composites

**DOI:** 10.3390/polym12112522

**Published:** 2020-10-29

**Authors:** Napisah Sapiai, Aidah Jumahat, Mohammad Jawaid, Anish Khan

**Affiliations:** 1Faculty of Mechanical Engineering, Universiti Teknologi MARA (UiTM), Shah Alam, Selangor 40450, Malaysia; napisah@uitm.edu.my; 2Institute for Infrastructure Engineering Sustainable and Management (IIESM), Universiti Teknologi MARA, Shah Alam, Selangor 40450, Malaysia; 3Department of Biocomposite Technology, Institute of Tropical Forestry and Forest Products, Universiti Putra Malaysia, UPM Serdang, Selangor 43400, Malaysia; 4Center of Excellence for Advanced Materials Research, King Abdulaziz University, P.O. Box. 80203, Jeddah 21589, Saudi Arabia; anishkhan97@gmail.com

**Keywords:** multiwall carbon nanotubes, acid treated, silane treated, kenaf fibres, glass fibres

## Abstract

The aim of this study is to evaluate the effect of surface treated multi wall carbon nanotubes (MWCNTs) on compressive properties of the unidirectional (UD) kenaf and hybrid woven glass/UD kenaf fibre reinforced polymer composites. The MWCNTs were first treated using concentrated acid (a mix of H_2_SO_4_ and HNO_3_) and silane (three-aminoprophyltriethoxysilane) in order to improve the dispersion within the epoxy matrix using a high shear roll milling technique. In this study, nanomodified epoxies were prepared using 0.5, 0.75 and 1.0 wt % of pristine MWCNT (PCNT), acid treated MWCNT (ACNT) and silane treated MWCNT (SCNT). These nanomodified epoxies were then used for the fabrication of kenaf and hybrid composites using combination of filament winding and resin impregnation. The uniaxial compression test was conducted using a universal testing machine according to the ASTM D3410 standard. The morphology of fractured samples was observed and analysed using scanning electron microscopy (SEM) in order to evaluate the failure behaviour and mechanisms involved during compression. It was found that the addition of treated MWCNT (ACNT and SCNT) improved the compressive properties of kenaf and hybrid composites as compared to those of untreated-MWCNT (PCNT). The addition of 1.0 wt % of SCNT exhibited good compressive properties in both kenaf and hybrid composite systems. The compressive modulus and strength increased by 73.25% and 20.15%, respectively, for composites made of 1.0 wt % SCNT and Kenaf (1.0SCNT/K). For the hybrid composites, the compressive modulus and strength increased by 21.18% and 7.73% for composites made of 1.0 wt % SCNT filled G/K composites (1.0SCNT/G/K).

## 1. Introduction

The concern of environmental pollution and the depletion of petroleum-based resources has attracted material science researchers to develop new eco-friendly green materials based on sustainability principles. Over the last two decades, the natural fibre reinforced polymer composites have been increasingly used in construction and automotive industries as substitutes for synthetic fibre reinforced polymer composites, especially glass and carbon fibres. The mechanical properties of natural fibre composites are comparable to those of synthetic fibres. Accordingly, the natural fibres have already proven that their capability can satisfy the requirement of the global market, especially for industries which are concerned about weight reduction. For this reason, numerous studies have investigated the properties and behaviour of kenaf fibre reinforced polymer composites to improve their performance and widen their scope of applications [1,2,3,4,5,6,7,8,9,10,11].

Kenaf fibres have been identified as an attractive option for reinforcing fibres due to their wide availability, high aspect ratio and superior toughness. Even though kenaf fibre reinforced polymer composites have good potential for commercialisation, certain limitations arise with respect to the mechanical properties when kenaf fibres and epoxy resin are used. The combination of these materials was reported to possess poor interfacial strength between the fibres and matrix due to the hydrophilicity of kenaf fibres and hydrophobicity of epoxy resin [2,11,12,13,14,15]. In addition, the majority of epoxy systems are characterised by brittleness, low impact strength and low fracture toughness once they are fully cured. Due to these limitations, further studies have been extensively conducted in order to overcome the problems. 

Generally, the use of nanofillers such as carbon nanotubes (CNTs), nanoclay, graphene, nanoalumina and nanosilica are commonly used for improving the properties of composite materials [9,10,16,17,18,19,20,21,22,23,24,25]. The discovery of CNTs has offered exciting opportunities in various applications of science and engineering. Both theoretical and experimental results suggest excellent properties of CNTs, which potentially can improve the physical, thermal, mechanical and electrical properties of polymer composites [16,26,27]. However, researchers are still facing challenges in using CNTs in the most effective way. The major challenges in utilizing CNT are the poor interfacial bonding between CNT and the polymer matrix, poor dispersion of the CNTs within the polymer matrix, improper alignment of the CNTs and degradation of the CNTs due to processing. Most researchers claimed that the agglomeration of CNTs due to the attraction of Van der Waals forces leads to the inefficiency of CNTs inclusion. Consequently, the predicted properties of the CNTs could not be fully reached in modified polymer nanocomposites. Regarding this issue, there were several surface modification methods suggested for improving the quality of the filler-matrix bonding and increasing the dispersion of CNTs in various matrices [16,28,29,30,31,32,33,34,35,36,37]. Ma et al. [38] discovered that the silane treatment was an effective method as it led to good dispersion of carbon nanotubes in ethanol without damaging its structures. The suspension stability of pristine MWCNTs was poor due to agglomeration and poor hydrogen-bonding ability as compared to the chemically modified MWCNT. Chemical modification of MWCNT results in the presence of functional groups or epoxy ended-groups that are covalently bonded on the surface of the MWCNTs. The presence of these functional groups modifies the surface characteristics of the MWCNTs and improves the dispersion of MWCNTs within epoxy resin.

This study will focus on the acid treatment using the mixture of H2SO4/HNO3 and silane functionalisation by 3-Aminopropyl Triethoxysilane to enhance dispersion of the MWCNTs.

In addition to modifying the epoxy resin with nanofillers, hybridisation of composites, which is the combination of two or more different fibres, also offers broader insight into the possibilities of improving the performance of composites. Satisfactory performance could be achieved when both of the hybridised fibres cooperate with each other to deliver good mechanical strength. Over the last few years, the hybridisation of natural fibres with synthetic fibres has received great attention [17,39,40,41,42,43,44,45]. The glass fibres have been used since they possess high mechanical strength with low cost as compared to other synthetic fibres. The development of the hybrid jute/glass fibres composites by Korodiya et al. [46] proved that these two fibres have their own specialty which contribute to the final composite performance. The study suggested that jute fibres had contributed to the improvement of toughness properties by promoting crack propagation, whereas glass fibres contributed to the improvement of thermal stability, water absorption behavior, overall strength and stiffness of the hybrid composites. Saidane et al. [42] studied on hybridisation effect of flax–glass fibres on kinetic diffusion and tensile properties. The authors observed that the Young’s modulus and tensile strength of flax fibres composites had been improved by the incorporation of glass fibres. This improvement was caused by the properties of glass fibres, which were comparatively stiffer than flax fibres. They also found that the hybridisation of flax and glass fibre composites improved the moisture resistance by reducing the water absorption and the diffusion coefficient. This hybrid flax-glass fibres composites did not have any negative effect on the tensile strength and specific tensile strength when exposed to wet environment at high temperatures.

From recent research as reported in the literature, it is shown that mechanical properties, especially compression properties, of CNTs as nanofillers have not been fully explored for natural fibre reinforced composites application. Therefore, the study of the CNT-modified kenaf and CNT-modified hybrid G/K composites, which are considered as environmentally friendly nanocomposites, is essential. The addition of CNTs is a significant approach to enhance the performance of kenaf composites, while its surface modification is needed to overcome the dispersion and adhesivity of the kenaf and hybrid G/K composites. The impact of the hybridization of kenaf with glass and the addition of CNTs in hybrid glass/kenaf composite systems on compression properties of nanocomposites is described in this work. The development of kenaf fibre based composites should be expanded in order to fully utilize the natural fibre resources in various applications.

## 2. Materials and Methods 

### 2.1. Materials

The materials used for the fabrication of kenaf and hybrid kenaf/glass fibre reinforced polymer composites are epoxy resin, MWCNT, kenaf fibres and glass fibres. The epoxy used (named miracast 1517 A/B) was purchased from the MIRACON Sdn. Bhd, Selangor, Malaysia. Miracast 1517 A/B is a DGEBA epoxy which was designed for laminate application. The amine-curing agent was used as a hardener with the ratio of 100:30 (epoxy: hardener). The Flo Tube 9000 Series MWCNT used was purchased from CNano Technology, Beijing, China. The yarn type of kenaf fibre and woven type of glass fibre were used as reinforcing materials. The yarn kenaf fibre, with an average diameter of 1.54 mm was supplied by Innovative Pultrusion Sdn Bhd., from Negeri Sembilan, Malaysia, while the woven glass fibre roving (CRW200) was supplied by Vistec Technology Sdn. Bhd., Selangor, Malaysia.

In the surface modification process of MWCNT, several chemicals were used such as nitric acid (HNO_3_), sulfuric acid (H_2_SO_4_), 3-aminopropyl triethoxysilane and ethanol. The nitric acid (HNO_3_), sulfuric acid (H_2_SO_4_) and ethanol were supplied by R & M Chemicals (Selangor, Malaysia) and 3-aminopropyl triethoxysilane was supplied by Sigma-Aldrich, Saint Louis, MO, USA.

### 2.2. Surface Modification of MWCNT

Surface modification is necessary for improving the dispersion of MWCNT in the epoxy matrix and to enhance the interfacial adhesion between MWCNT and the epoxy matrix. The surface modification aims to attach the functional groups onto the MWCNT surfaces. In the acid treatment process, 3.0 g of pristine MWCNT was dispersed in a flask containing 300 mL of acid solution. The acid solution was prepared with a ratio of 3:1 of sulfuric acid (H_2_SO_4_) and nitric acid (HNO_3_). The mixture of MWCNT and acid solution was then homogenously mixed and refluxed for 4 h at 80 °C using a combined hot-plate magnetic stirrer and reflux device. After 4 h of the refluxed process, the mixture solution was poured into a 2000 mL beaker before distilled water was added into the solution. The mixture was continuously stirred for 6 h to ensure a uniform dispersion of the CNT. The mixture was then filtered and washed with distilled water and acetone until a pH value of 6–7 was reached, to remove the remaining acid. The acid treated-CNT (ACNT) was then dried in the oven for 80 °C for 24 h. The dried ACNT was ground by agate mortar and planetary ball milling for 30 min.

In the silane functionalisation process, 1.8 g ACNT powders were dispersed in 300 mL of an ethanol–water–silane solution. The solution was diluted with 2 wt % of three-aminoprophyltriethoxysilane to an aqueous solution of 300 mL ethanol–water solution with a ratio of 95:5 (vol.% ethanol: vol.% water). The mixture was stirred at 70 °C for 4 h to facilitate the reaction between active sites (–OH group) of the ACNT and NH_2_–(CH_3_)–Si (OH)_3_. The saline treated-CNT (SCNT) was next separated by a filtration process and washed several times using acetone and distilled water to remove the remaining silane. The filtered SCNT was dried at 80 °C for 12 h. The processes were continued by grinding and milling to get a fine powder of SCNT.

### 2.3. Fabrication of Unidirectional (UD) Kenaf and Hybrid UD Kenaf/Glass Fibres Reinforced Polymer Composites

In this study, eight systems of kenaf fibre reinforced composite samples were prepared: unidirectional (UD) kenaf (K) composite, hybrid G/K composite, pristine MWCNT (PCNT)-modified kenaf composites, ACNT-modified kenaf composites, SCNT-modified kenaf composites, PCNT-modified hybrid G/K composites, ACNT-modified hybrid G/K composites and SCNT-modified hybrid G/K composites. Glass fibre (G) composite samples were prepared in order to use as a reference material. The formation of these composites involved a filament winding process to align yarn kenaf fibres unidirectionally; then, the resin was impregnated into wound kenaf fibres. For the hybrid systems, the woven glass fibres were placed on top and underneath the wound kenaf fibres, before being impregnated with epoxy resin. The impregnated kenaf and hybrid G/K were left for curing for 24 h at room temperature. Finally, the post curing process was done to improve the thermal properties of the kenaf composites. The post curing cycle for epoxy was suggested by the epoxy manufacturer as illustrated in Figure 1. The post curing was started at 60 °C for 2 h, followed by 80 °C for 2 h, 100 °C for 2 h and then 120 °C for another 2 h. Finally, the composites were cut into specific size according to the compression test standards. 

The series of CNT-modified kenaf and CNT hybrid G/K composite systems were prepared using similar steps as the kenaf and hybrid G/K composites. Prior to that, the series of CN-modified epoxy resin needed to be prepared. First, the CNT (PCNT/ACNT/SCNT) powder was weighed at three different weight percentages which were 0.5, 0.75 and 1.0 wt % of CNT. After that, the powder was put into a flask containing acetone. A sonication process was next performed for 1 h, before mixing the result with epoxy resin. The mixture of CNT and epoxy resin was then left for 24 h to evaporate the remaining acetone. The mixing process was then continued using high shear roll milling for 1 h at 400 rpm. The CNT-modified epoxy resin was impregnated into wound kenaf for fabrication of the CNT-modified kenaf and hybrid G/K composites. The process was followed by post curing before the composites were ready to be characterised.

### 2.4. Designation of Kenaf and Hybrid Kenaf/Glass Fibres Reinforced Polymer Composites

Surface functionalization was performed on the MWCNT surfaces as coded as ACNT for acid treated MWCNT and SCNT for silane treated MWCNT; as-received MWCNT was coded as PCNT. The formulation of the kenaf, hybrid, and CNT modified composites fabricated in this work shown in Table 1.

### 2.5. Characterization

#### 2.5.1. Dispersion Evaluation and Microstructure Characterization

Transmission electron microscopy (TEM), model FEI Tecnai G2 20S TWIN was used to examine the dispersion of the PCNT, ACNT and SCNT within the epoxy matrix. The PCNT, ACNT and SCNT modified with epoxy resin were cured into trapezium shape specimens and then ultra-thin sections (thickness of 85 nm) were prepared using a Leica Ultra Microtome machine, before placing them on a copper grid. The TEM was carried out at Hospital Universiti Kebangsaan Malaysia (HUKM).

A Hitachi TM 3000 SEM was employed to examine the fractured surface of the kenaf and hybrid kenaf/glass fibre reinforced polymer composites after they were exposed to the compression tests. The observation of the fractured surface is important to determine the failure mode that occurred during the compressive test. The SEM image was also used to observe the quality of the fibre–matrix adhesion in the composite system. The fractured specimens were stacked on aluminium stubs with carbon adhesive tape and coated with a thin layer of gold in order to prevent electrostatic charging during observation. Practically, a coating of gold was applied to provide a conducting film and to enhance the electron emissions to the surface area of the specimens. The specimens were then placed in the specimen chamber under vacuum conditions at an accelerating voltage of 5 KV. The SEM was carried out at the Faculty of Dentistry, UiTM Sungai Buluh Selangor.

#### 2.5.2. Compression Test

In this study, the compression test was conducted in accordance with the ASTM D3410 standard. The compression test was conducted using an Instron universal tester machine with a special rig design and the test results were analysed using BlueHill data software as shown in Figure 2. The special fixture test rig was designed to support the specimen, as well as to prevent the specimen from buckling. Compression tests were conducted on unidirectional laminates to determine the compressive stress, compressive strain and compressive modulus. The compressive modulus of the composite samples was calculated at 1% to 3% compressive strain. The sample with a size of 110 mm × 10 mm × 5 mm, as illustrated in Table 2, was used for this compressive test. At least five identical specimens for each composite system were tested to determine their properties. The compressive stress and compressive strain were evaluated using Equations (1) and (2), respectively, as follows:(1)$$\sigma \left(MPa\right)=\frac{F}{A}$$
(2)$$\varepsilon \;\left(\%\right)=\frac{\Delta l}{l}x\;100$$
where *F* = the force at the crack extension, *A* = the area of the samples (width x thickness), Δl = the extension of the sample and l = the gauge length of the sample.

## 3. Results and Discussion

### 3.1. TEM Evaluation for Dispersion of MWCNT (PCNT/ACNT/SCNT) within Epoxy Matrix

The use of MWCNT as nano-filler is being considered in order to improve the mechanical performance of the reinforced polymer composites. Since it was discovered by Ijima in 1992, MWCNT had an exceptional nano-scale physical and mechanical properties. These include having a Young modulus of 1–5 TPa, high elongation to failure in the range 20–30%, excellent thermal stability and low electrical resistance. However, these good properties of MWCNT cannot be utilized due to poor dispersion of MWCNT within the polymer matrix, poor interfacial adhesion between the MWCNTs and the matrix, improper MWCNT alignment and degradation of the CNT during processing [30,34,47]. The surface functionalisation of the CNT surface has been regarded as an effective technique in improving MWCNT-matrix interfacial bonding strength and increasing the dispersion of the matrix in various polymer composites. The dispersion of PCNT, ACNT and SCNT within epoxy resin are illustrated in Figure 3, Figure 4 and Figure 5, respectively. Figure 3 illustrates the entangled cluster PCNT within the epoxy matrix. With high content of PCNT, the bigger clump of entangled PCNTs was revealed. The existence of the bigger clump of entangled PCNT is due to high viscosity of epoxy resin when increasing the amount of PCNT. This phenomenon eventually will lead to ineffectiveness of stress transfer, thus leading to poor mechanical properties of the composites.

The acid treatment process on MWCNT surfaces was found to improve the dispersion of ACNT within the epoxy matrix as shown in Figure 4. It was suggested that the bonding between ACNT and epoxy matrix was improved by the bonding of functional group –COOH on ACNT surfaces with molecules of epoxy resin. The dispersion of ACNT was much better compared to PCNT in the epoxy matrix.

The dispersion of the 0.5, 0.75 and 1.0 wt % of SCNT within the epoxy matrix is illustrated in Figure 5. The TEM images show a good distribution of SCNT within the epoxy matrix even at a high content of SCNT (1.0 wt %). As reported from other researchers, the agglomerated structures of MWCNT occurred due to Van der Waals interaction between nanotubes [19,47,48,49,50,51,52]. With the existence of the hydroxyl group (–OH) to form a covalent Si–O–Si bond after the salinization process, this could be an effective way to improve the dispersion of MWCNT within epoxy resin.

### 3.2. Effect of Hybridization on Compressive Properties of Kenaf Fibres Reinforced Polymer Composites and Kenaf/Glass Hybrid Composites

In this study, with the aim of improving the compressive properties of kenaf composites, the hybridisation with glass fibres was suggested. From the results obtained, both kenaf (K) and hybrid G/K composites possessed lower mechanical properties when compared to glass (G) composite. The increment in mechanical properties of hybrid G/K was due to the fact that glass fibres are stronger, stiffer and more extensible than kenaf fibres. Indeed, in the case of hybridisation, the composites mainly depend on mechanical characteristic of the individual reinforcing fibres [42]. As reported in the literature, there are few factors contributing to the mechanical properties of hybrid composites such as fibre volume fractions and stacking sequence i.e., the pattern of arrangement of the fibre layers in hybrid composites. In this study, the fibre volume fraction of glass fibre in the hybrid composite system (G/K) is small of about 5–7 vol. % when compared to the pure glass composite system (G) of 47 to 50 vol.%. Therefore, the presence of small volume fraction of glass fibre at outside layers of the Hybrid G/K composites slightly improve the Young’s Modulus and strength. This hybridisation clearly shows the complementary potential between natural and synthetic fibres.

Based on the observation during the experimental session, the micro buckling and delamination of glass fibres occurred on the glass (G) and the hybrid G/K composites when subjected to compressive loading. The effect of compressive loading and the fracture mechanisms involved were captured and supported via typical compressive stress–strain curves as shown in Figure 6, in which the curves (refer to the glass (G) and the hybrid G/K composites) have a bilinear line due to premature failure of the brittle glass fibres. As a result, glass (G) composite indicated a lower compressive strength than was expected. The compressive strength of hybrid G/K composite indicated very small improvement compared to that of kenaf (K) composite. As shown in Figure 7, micro-buckling failure in shear mode was observed in these fractured surfaces of composites. The SEM images of fractured surfaces of kenaf (K) and hybrid G/K composites displayed fibre waviness, matrix cracking and debonding of fibres, as can be seen in Figure 8. The shear mode failure occurred at high fibre volume fractions and created a shear strain in the matrix because of in- phase buckling of the fibres [53]. Table 3 shows the compressive strength of the kenaf (K), glass (G) and hybrid glass kenaf composites. The results recorded are about 68.78 ± 1.08, 74.34 ± 0.41 and 73.24 ± 0.77 MPa for kenaf (K), glass (G) and hybrid glass kenaf composites, respectively. The compressive modulus increased when hybridised with glass fibres compared to kenaf (K) composite. The compressive modulus was verified with the values of 2.43 ± 0.17, 6.06 ± 0.16 and 3.57 ± 0.14 GPa for kenaf (K), glass (G) and hybrid G/K composites, respectively. As reported from other researchers, there are few factors that contributed to the enhancement of compressive properties such as the fibre type, fibre volume fraction, matrix yield strength, fibre length-diameter ratio, fibre straightness, and fibre alignment, as well as the fibre-matrix interfacial shear strength [45,54,55].

### 3.3. Effect of Surface Functionalisation MWCNT on Compressive Properties of Kenaf Fibres Reinforced Polymer Composites

Figure 9a shows the compressive stress-strain behaviour of PCNT-modified kenaf composites, ACNT-modified kenaf composites and SCNT-modified kenaf composites. The measured compressive modulus was found to increase with an increasing amount of PCNT. The incorporation of 0.5, 0.75 and 1.0 wt % PCNT had enhanced the compressive modulus by 0.82%, 12.76% and 18.93%, respectively corresponding to the kenaf (K) composite. The compressive strength for 0.5 and 0.75 and 1.0 wt % PCNT- modified kenaf composites increased by 6.8%, 9.3% and 6.2%, respectively, when compared to those of kenaf (K) composite. However, the inclusion of 1.0 wt % PCNT had decreased the compressive strength as compared to those with the inclusion of 0.5 and 0.75 wt % PCNT. It is believed that the interface between the three phases of CNT, epoxy matrix and kenaf fibres had influenced the compressive properties of the PCNT- modified kenaf composites. The inclusion of PCNT had enhanced the compressive properties. However, due to the poor dispersion and agglomerated structure as shown in Figure 3, in the 1.0 PCNT- matrix region, the compressive strength started to degrade. The agglomerated structure also leads to poor adhesion bonding between the CNTs and the epoxy matrix.

The effect of ACNT on the compressive strain–strain response of ACNT-modified kenaf composite is illustrated in Figure 9b. The measured modulus was found to increase with an increasing amount of ACNT. The incorporation of 0.5, 0.75 and 1.0 wt % ACNT enhanced the compressive modulus by 34.98%, 51.85% and 56.38%, respectively, while the compressive strength for 0.5 and 0.75 and 1.0 wt % PCNT- modified composites increased by 8.55%, 12.13% and 14.10%, respectively, when compared to kenaf (K) composite. It can be seen that the presence of ACNT enhanced the compressive stress–strain behaviour of the ACNT-modified kenaf composites. The presence of ACNT gave higher resistance against deformation, and thereby resulted in improved compressive strength. The improvement on compressive properties also implied that the acid modification enhances the dispersion of ACNT within the epoxy matrix, thus repairing the three-phase adhesion bonding of ACNT-kenaf modified composites.

The effect of SCNT on the compressive stress–strain response of SCNT-modified kenaf composites is illustrated in Figure 9c. It can be seen that the addition of SCNT enhanced the compressive properties of the kenaf composites. The slopes of the compressive stress–strain curves increased with increasing SCNT, proving the enhancement of stiffness when SCNT was incorporated. Table 4 summarises the compressive properties of SCNT-modified kenaf composites compared to kenaf (K) composites. The addition of 0.5, 0.75 and 1.0 wt % of SCNT enhanced the compressive modulus by 43.21%, 67.49% and 73.25%, respectively, and the compressive strength by 12.70%, 16.16% and 20.15%, respectively. As reported earlier, the CNT with surface modification had proven to improve the dispersion of CNT within the epoxy matrix and enhanced three-phase bonding between CNT, epoxy matrix and kenaf fibres. The strong interfaces bonding had contributed to the effectiveness of the stress transfers capability, thus enhancing the compressive modulus and strength of the composites. This phenomenon can be explained in term of attachment of the functional group on CNT after surface modification. Silane molecules with epoxy end-groups were covalently attached on the CNT surface via the silanisation reaction and partial hydrolysis of silane after functionalization. The ACNT consumed to have O–H and C=O functional groups, whereas SCNT consumed to have Si–OH, Si–O–Si and C=0 functional groups. These functional groups reacted with molecules of epoxy resin to create covalent bonding between the CNT and epoxy matrix, and this may increase the stiffness of the composites. A similar finding was also reported by Lavorgna et al. and Vennerberg et al. in their research on silanized MWCNT-epoxy nanocomposites [28,47].

It is interesting to note that after being subjected to a compression test, the SEM image of kenaf (K) composite displayed the biggest crack compared to CNT-modified kenaf composites, as depicted in Figure 10. The micro spot cracks observed in the SEM image of 0.5SCNT/K indicate that the strong SCNT-epoxy bonding of the surface modification of CNT (SCNT) was effective in improving the quality of the CNT-modified kenaf composites. This finding also supports the conclusion that the compressive properties of kenaf composites depend more on the epoxy matrix rather than the kenaf fibres. It is difficult to determine kenaf fibres buckling/failure after the matrix crashed and cracked. The kenaf composites completely failed when the matrix failed. The compressive properties for PCNT-modified kenaf composites, ACNT-modified kenaf composites and SCNT-modified kenaf composites are summarised in Table 4.

### 3.4. Effect of Surface Functionalization MWCNT on Compressive Properties of Hybrid Kenaf/Glass Fibres Reinforced Polymer Composites

Figure 11a illustrates the typical compressive stress–strain curves of hybrid G/K, 0.5PCNT/G/K, 0.75PCNT/G/K and 1.0PCNT/G/K composites. It was observed that the compressive stress–strain curves have a bi-linear pattern. When the kenaf composites were hybridised with glass fibres, the curves initially started with a linear line and the second linear line occurred at a compressive strain of 10–15%. In the hybrid system, the specimens failed prematurely due to micro-buckling, which was triggered by longitudinal splitting of fibre glass. This failure was also due to the incompatibility i.e., weak bonding between glass fibres with the epoxy matrix and kenaf fibres. However, there are many factors, as discussed earlier, which may have affected the performance of the composites, especially when using natural fibres as a reinforcement. In regard to compressive properties, the inclusion of PCNT improved the compressive modulus by 6.44%, 14.29% and 14.57% for 0.5PCNT/G/K, 0.75PCNT/K/G and 1.0PCNT/K/G, respectively, compared to hybrid G/K, while the compressive strength recorded was the within range of 73.24–74.26 MPa. Even though there was an agglomerated structure CNT region, which is expected to give detrimental effect to the composites, the compressive properties can be retained with glass fibre support. As compared to the kenaf composites system, the hybridisation with glass fibres increased the compressive modulus but did not improve compressive strength. This was due to micro-buckling i.e., premature failure, as supported by the compressive stress–strain curves in Figure 11a.

The compressive stress–strain behaviour of ACNT-modified hybrid G/K composites is shown in Figure 11b. The hybridisation with glass suffered from micro-bucking, thus leading to the delamination of the glass fibres. This finding was supported by the compressive stress–strain graphs, which can be interpreted to mean that the second linear lines happened after delamination of the glass fibres. The graph behaviour also showed poor interface bonding between glass fibres with kenaf composites. As compared to kenaf (K) composite, the inclusion of the ACNT slightly improved the compressive modulus and strength. The compressive moduli were 3.95 ± 0.09, 4.21 ± 0.04 and 4.23 ± 0.11GPa and the compressive strengths were 75.05 ± 0.59, 75.23 ± 0.82 and 76.06 ± 1.41 MPa for 0.5ACNT/G/K, 0.75ACNT/G/K and 1.0ACNT/G/K composites, respectively. 

The effect of SCNT on the compressive stress–strain response of the hybrid G/K composites is illustrated in Figure 11c. The graph indicates that the addition of SCNT slightly improved the compressive properties. The addition of 0.5SCNT, 0.75SCNT and 1.0SCNT enhanced the compressive modulus by 17.09%, 19.93% and 21.18% and increased the compressive strength by 3.45%, 5.09% and 7.73%, respectively. As discussed earlier, CNT with surface modification have proven to improve the dispersion of CNT within the epoxy matrix and enhanced the three-phase bonding between CNT, epoxy and the fibres. Strong interfaces bonding was believed to contribute to the effectiveness of the stress transfer capability, thus enhancing the compressive modulus and strength of the composites. However, in the hybrid G/K composites system, the addition of CNT did not show the good effect as compared with kenaf composite systems. This was due to premature failure i.e., micro-buckling when hybridised with glass fibres. The premature failure was again supported by compressive stress–strain graphs which have bi-linear behaviour, and similar results were found in (a) PCNT-modified hybrid G/K composites and ACNT-modified hybrid G/K composites. The SEM images of the fractured surface after subjected to compression loading shows that the hybrid composites failed with brittle failure of the glass fibres, as illustrated in Figure 12. The existence of the glass fibres gives negative impact to the compressive performance due to the brittleness and poor bonding between the glass fibres and epoxy resin. Table 5 summarises the compressive properties of hybrid G/K composite, PCNT-modified hybrid G/K composites, ACNT-modified hybrid G/K composites and the SCNT-modified hybrid G/K.

## 4. Conclusions

It is concluded that the addition of PCNT in kenaf composites shows a reduction in compressive properties due to the existence of an agglomerated region within the epoxy matrix. The inclusion of ACNT and SCNT significantly improved the compressive properties of CNT-modified kenaf composite systems. It is concluded that the silane functionalisation improves the dispersion and bonding between the epoxy matrix, that offset the inferiority of the compressive properties for kenaf composites. The functionalized CNT contributes to a better performance of kenaf fibre composites due to three main factors: (i) good dispersion of functionalised CNT in epoxy, (ii) better load transfer mechanisms occur in between the matrix and the fillers due to the formation of covalent bonds and (iii) improvement on the interfacial adhesion between the epoxy matrix and the CNT.

## Figures and Tables

**Figure 1 polymers-12-02522-f001:**
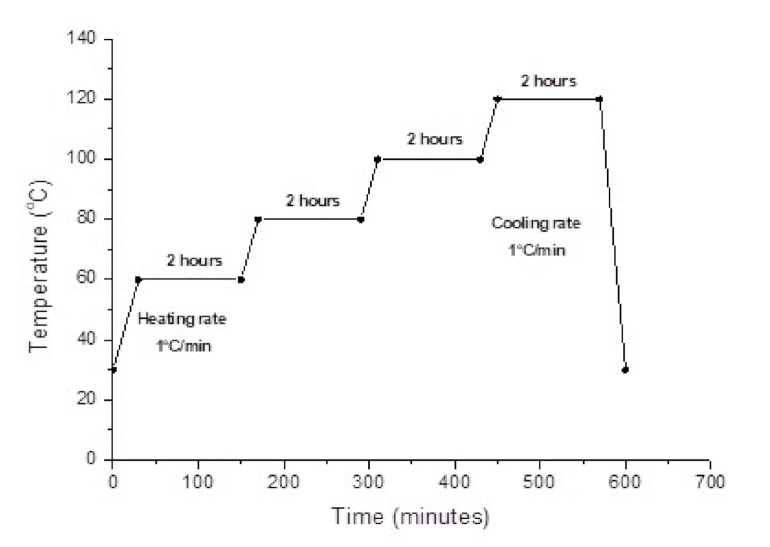
Post curing for the Miracast 1517A/B resin system (recommended by the manufacturer).

**Figure 2 polymers-12-02522-f002:**
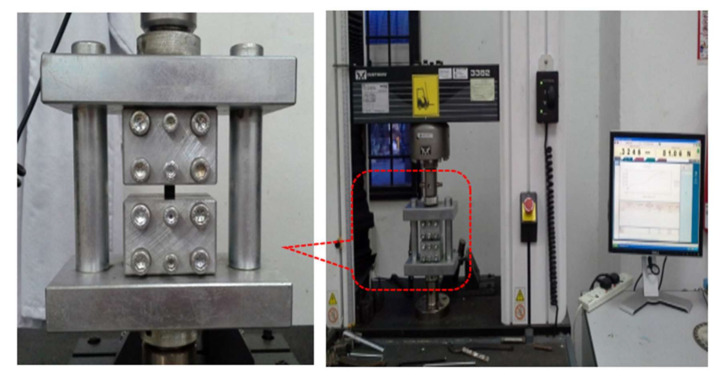
Instron universal testing machine with special rig design.

**Figure 3 polymers-12-02522-f003:**
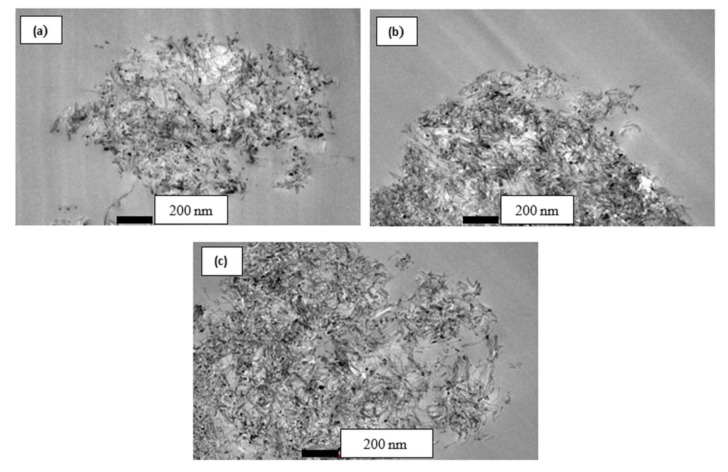
TEM micrograph showing a dispersion of (**a**) 0.5 wt %, (**b**) 0.75 wt % and (**c**) 1.0 wt % of PCNT in epoxy resin, observed under a magnification of 43,000×.

**Figure 4 polymers-12-02522-f004:**
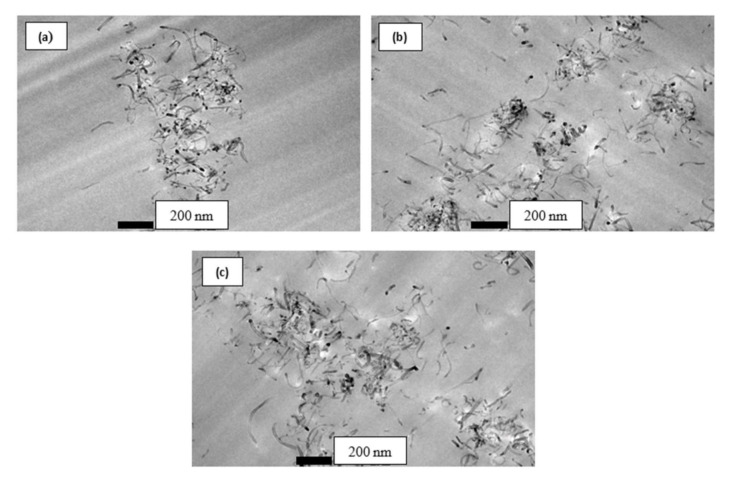
TEM micrographs showing a dispersion of (**a**) 0.5 wt %, (**b**) 0.75 wt % and (**c**) 1.0 wt % of ACNT in epoxy resin, observed under magnification of 43,000×.

**Figure 5 polymers-12-02522-f005:**
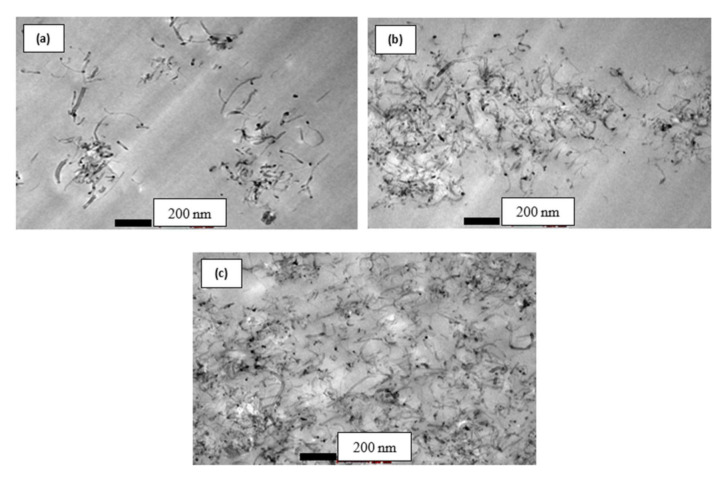
TEM micrographs showing a dispersion of (**a**) 0.5 wt %, (**b**) 0.75 wt % and (**c**) 1.0 wt % of SCNT in epoxy resin, observed under magnification of 43,000×.

**Figure 6 polymers-12-02522-f006:**
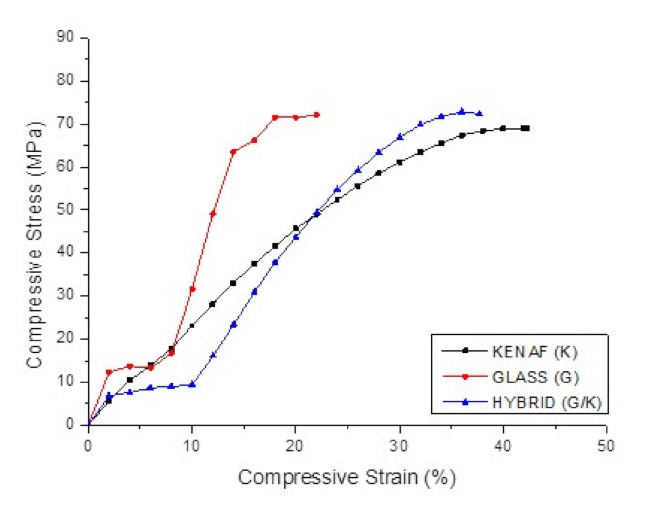
Typical compressive stress–strain curves of kenaf (K), glass (G) and hybrid G/K composites.

**Figure 7 polymers-12-02522-f007:**
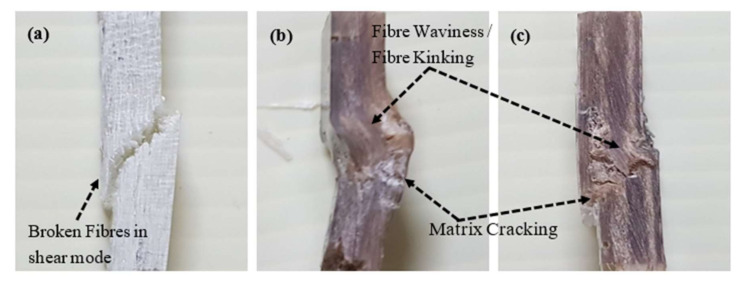
Samples of (**a**) glass (G) (**b**) kenaf (K) and (**c**) hybrid G/K composites after they were subjected to compressive loading.

**Figure 8 polymers-12-02522-f008:**
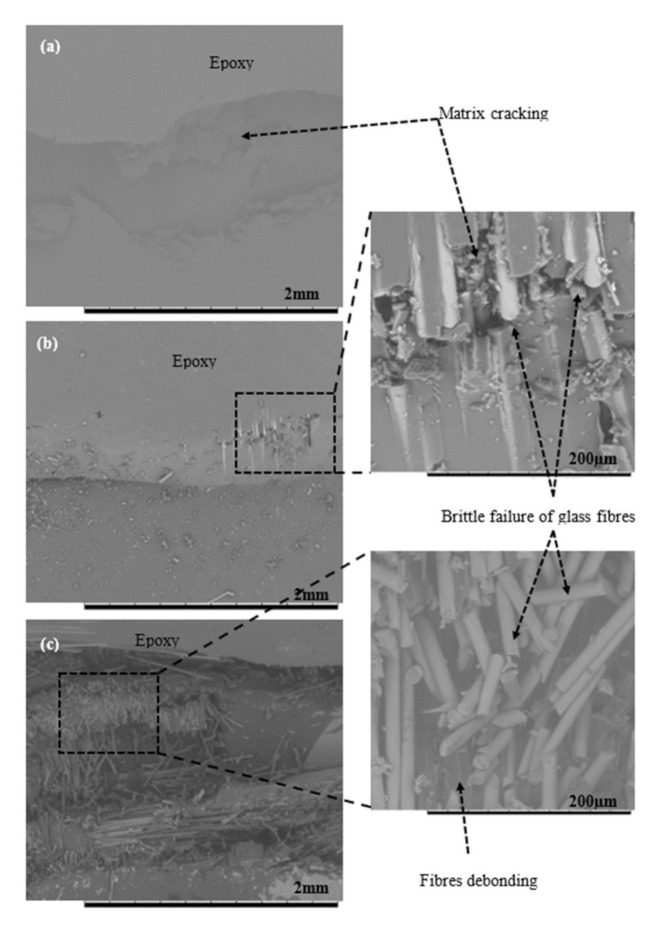
SEM images compressive fractured surfaces of (**a**) glass (G), (**b**) kenaf (K) and (**c**) hybrid G/K composites, magnification of 50× and 500×.

**Figure 9 polymers-12-02522-f009:**
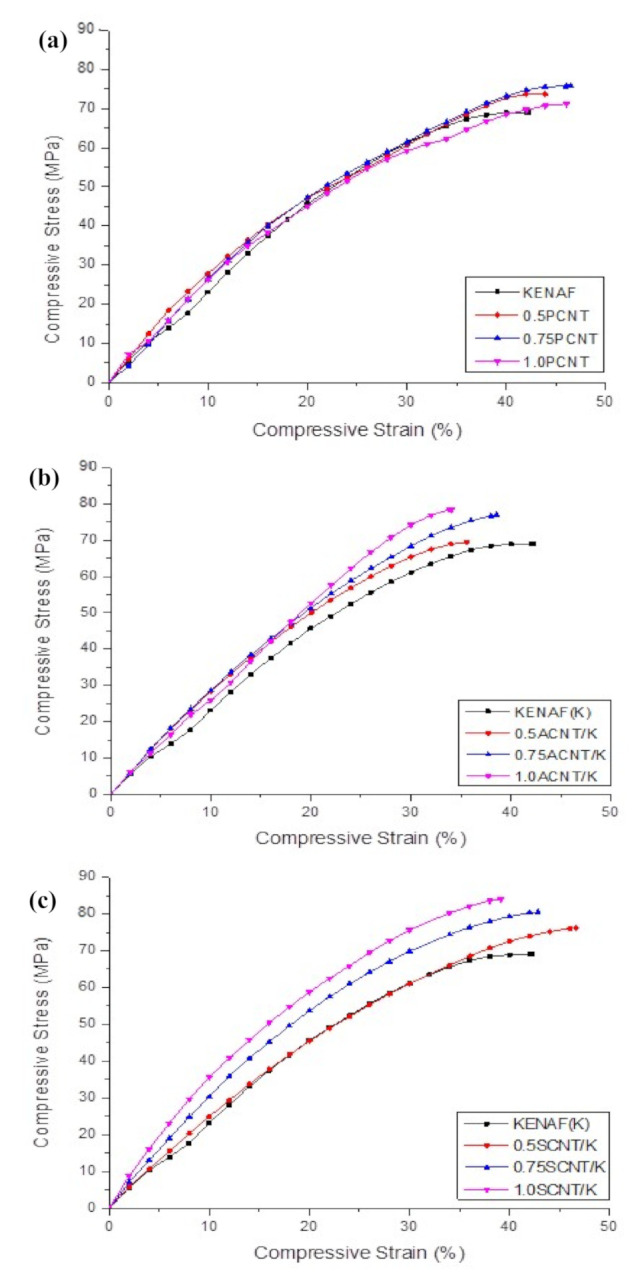
Typical compressive stress–strain curves of (**a**) PCNT-modified kenaf composites, (**b**) ACNT-modified kenaf composites and (**c**) SCNT-modified kenaf composites, as compared to kenaf (K) composites.

**Figure 10 polymers-12-02522-f010:**
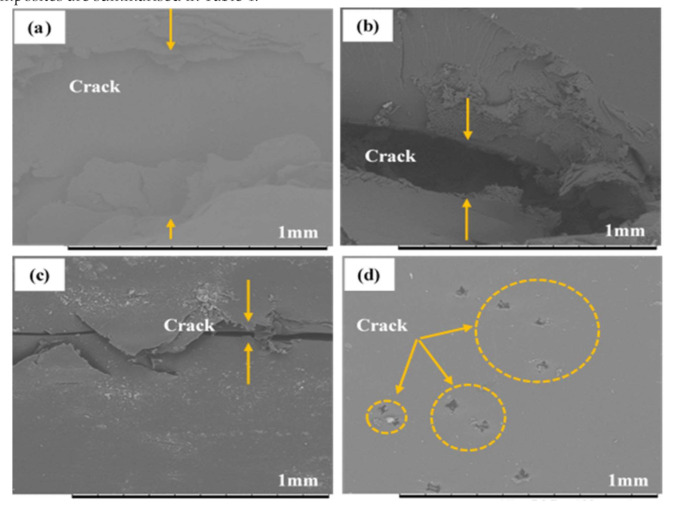
Matrix cracking observation in (**a**) kenaf (K) (**b**) 0.5PCNT/K (**c**) 0.5ACNT/K and (**d**) 0.5SCNT/K composites, after subjected to compression test.

**Figure 11 polymers-12-02522-f011:**
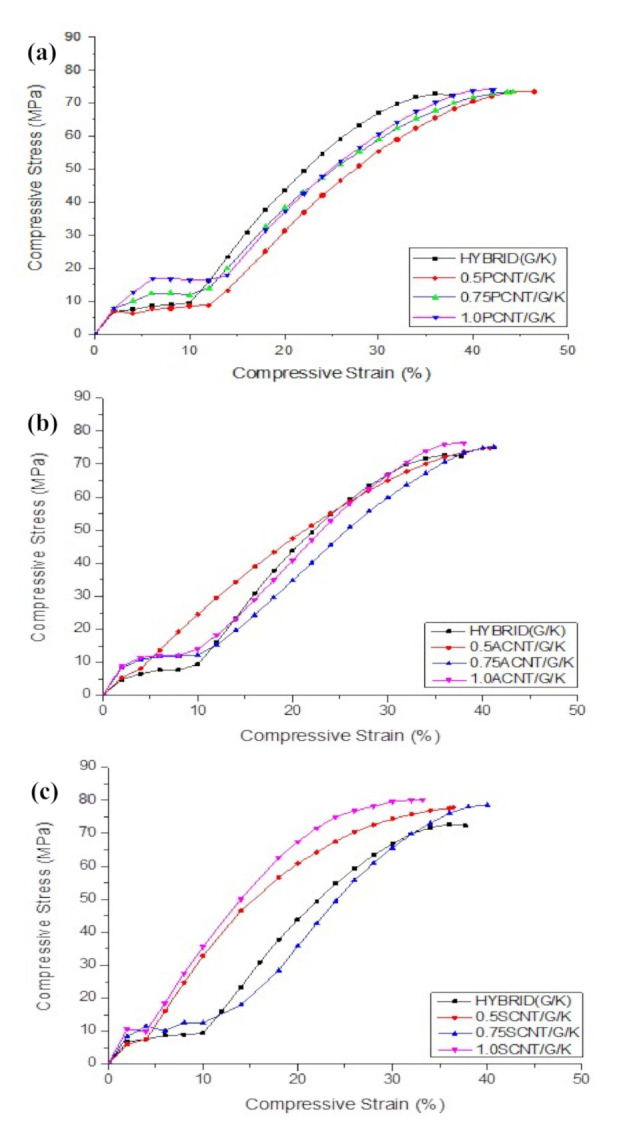
Typical compressive stress–strain curves of (**a**) PCNT-modified hybrid G/K composites, (**b**) ACNT-modified hybrid G/K composites and (**c**) SCNT-modified hybrid G/K composites, as compared to hybrid G/K composites.

**Figure 12 polymers-12-02522-f012:**
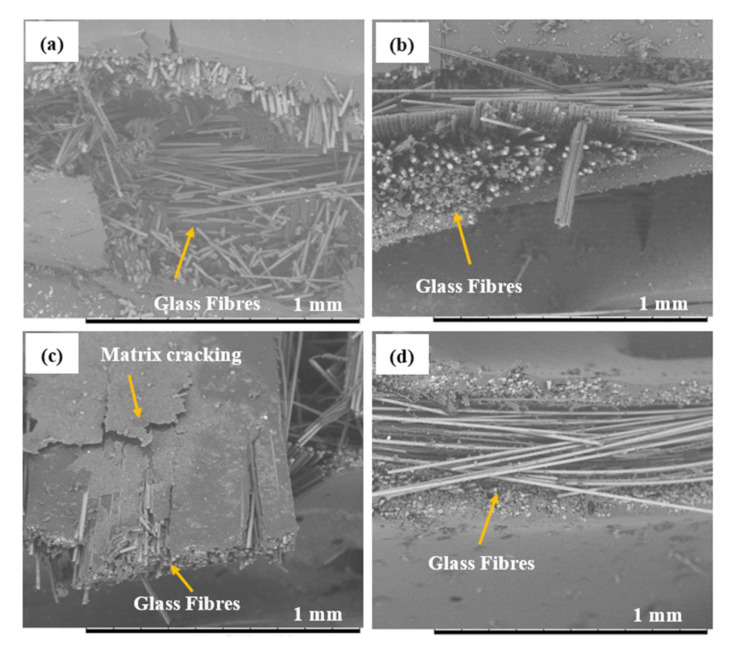
Matrix cracking and broken fibres observation: (**a**) hybrid G/K (**b**) 0.5PCNT/G/K (**c**) 0.5ACNT/G/K and (**d**) 0.5SCNT/G/K, after a compression test.

**Table 1 polymers-12-02522-t001:** The designation of kenaf and hybrid glass/kenaf composite systems.

Composite Systems	Designation
Unidirectional (UD) Kenaf Fibre Reinforced Polymer Composite	UD Kenaf (K)
Woven Glass Fibre Reinforced Polymer Composite	Woven Glass (G)
Hybrid Kenaf/Glass Fibre Reinforced Polymer Composite	Hybrid G/K
0.5 wt % PCNT modified Kenaf FRP Composite	0.5PCNT/K
0.75 wt % PCNT modified Kenaf FRP Composite	0.75PCNT/K
1.0 wt % PCNT modified Kenaf FRP Composite	1.0PCNT/K
0.5 wt % ACNT modified Kenaf FRP Composite	0.5ACNT/K
0.75 wt % ACNT modified Kenaf FRP Composite	0.75ACNT/K
1.0 wt % ACNT modified Kenaf FRP Composite	1.0ACNT/K
0.5 wt % SCNT modified Kenaf FRP Composite	0.5SCNT/K
0.75 wt % SCNT modified Kenaf FRP Composite	0.7SCNT/K
1.0 wt % SCNT modified Kenaf FRP Composite	1.0SCNT/K
0.5 wt % PCNT modified Hybrid Glass/Kenaf FRP Composite	0.5PCNT/G/K
0.75 wt % PCNT modified Hybrid Glass/Kenaf FRP Composite	0.75PCNT/G/K
1.0 wt % PCNT modified Hybrid Glass/Kenaf FRP Composite	1.0PCNT/ G/K
0.5 wt % ACNT modified Hybrid Glass/Kenaf FRP Composite	0.5ACNT/G/K
0.75 wt % ACNT modified Hybrid Glass/Kenaf FRP Composite	0.75ACNT/G/K
1.0 wt % ACNT modified Hybrid Glass/Kenaf FRP Composite	1.0ACNT/G/K
0.5 wt % SCNT modified Hybrid Glass/Kenaf FRP Composite	0.5SCNT/G/K
0.75 wt % SCNT modified Hybrid Glass/Kenaf FRP Composite	0.75SCNT/G/K
1.0 wt % SCNT modified Hybrid Glass/Kenaf FRP Composite	1.0SCNT/G/K

**Table 2 polymers-12-02522-t002:** The sample geometry for the compressive test.

Fibres Orientation	Width (mm)	Gage Length (mm)	Tab Length (mm)	Overall Length (mm)	Tab Thickness (mm)
0° unidirectional (UD)	10	10	50	110	1.5
	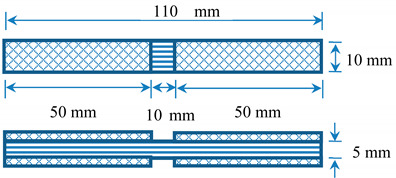				

**Table 3 polymers-12-02522-t003:** Compressive properties of kenaf (K), glass (G) and hybrid G/K composites.

Composites	Compressive Modulus (GPa)	Compressive Strength (MPa)	Compressive Strain at Break (%)
KENAF (K)	2.43 ± 0.17	68.78 ± 1.08	38.6 ± 4.91
GLASS (G)	6.06 ± 0.16	74.34 ± 0.41	19.92 ± 1.12
HYBRID (G/K)	3.57 ± 0.14	73.24 ± 0.77	37.13 ± 10.49

**Table 4 polymers-12-02522-t004:** Compressive properties of kenaf (K) composite, PCNT-modified kenaf composites, ACNT-modified kenaf composites and SCNT-modified kenaf composites.

Composites	Compressive Modulus (GPa)	Compressive Strength (MPa)	Compressive Strain at Break (%)
KENAF (K)	2.43 ± 0.17	68.78 ± 1.08	38.6 ± 4.91
0.5PCNT/K	2.45 ± 0.09	73.44 ± 2.18	41.84 ± 5.58
0.75PCNT/K	2.74 ± 0.20	75.17 ± 2.81	45.15 ± 3.71
1.0PCNT/K	2.89 ± 0.17	73.06 ± 2.20	48.20 ± 2.92
0.5ACNT/K	3.28 ± 0.19	74.66 ± 5.59	35.11 ± 3.11
0.75ACNT/K	3.69 ± 0.19	77.12 ± 1.29	40.39 ± 2.31
1.0ACNT/K	3.80 ± 0.06	78.48 ± 1.79	39.34 ± 3.02
0.5SCNT/K	3.85 ± 0.05	77.51 ± 2.54	47.54 ± 7.06
0.75SCNT/K	4.07 ± 0.14	80.09 ± 3.03	43.61 ± 5.66
1.0SCNT/K	4.21 ± 0.10	82.64 ± 2.16	38.12 ± 1.75

**Table 5 polymers-12-02522-t005:** Compressive properties of the hybrid G/K composite, PCNT-modified hybrid G/K composites, ACNT-modified hybrid G/K composites and SCNT-modified hybrid G/K.

Composites	Compressive Modulus (GPa)	Compressive Strength (MPa)	Compressive Strain at Break (%)
HYBRID (G/K)	3.57 ± 0.14	73.24 ± 0.77	37.13 ± 10.49
0.5PCNT/G/K	3.80 ± 0.20	73.55 ± 2.07	47.66 ± 8.95
0.75PCNT/G/K	4.08 ± 0.11	73.60 ± 1.87	44.59 ± 9.63
1.0PCNT/G/K	4.09 ± 0.07	74.26 ± 1.87	45.47± 3.65
0.5ACNT/G/K	3.95 ± 0.09	75.05 ± 0.59	41.21 ± 8.64
0.75ACNT/G/K	4.21 ± 0.04	75.23 ± 0.82	42.61 ± 6.98
1.0ACNT/G/K	4.23 ± 0.11	76.06 ± 1.41	39.22 ± 4.23
0.5SCNT/G/K	4.18 ± 0.10	75.76 ± 1.98	35.05 ± 2.38
0.75SCNT/G/K	4.26 ± 0.11	76.97 ± 2.56	44.46 ± 5.29
1.0SCNT/G/K	4.35 ± 0.05	78.90 ± 1.15	36.60 ± 3.14

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
