# Peer review of "Effect of MWCNT Surface Functionalisation and Distribution on Compressive Properties of Kenaf and Hybrid Kenaf/Glass Fibres Reinforced Polymer Composites"

_polymers, 2020, doi:10.3390/polym12112522_

Round 1

Reviewer 1 Report

This paper presents an interesting study of pristine and  silane functionalised  MWCNTs as reinforcing agents of  hybrid fibre material composites. 

There is high practical importance of jute, kenaf and other textile material  application studies. 

In the last decade, several studies on the application of MWCNTs and other carbonaceous materials have been reported, including composites based on kenaf and other polymers.

It would be recommended to better explain the innovation of this study comparing with the other reported studies. 

While 3-aminopropyltriethoxysilane is well known  coupling silane agent, this study presents interesting data of  its application for the improvement of  MWCNT dispersion in  kenaf/glass fibre composite system. 

The composites have been analysed at wide ratios of pristine, acid treated and  silane coupled   CNTS (0-1 wt.%) and  all the necessary experimental conditions for the surface, structure analysis and evaluation of stress t=strain characteristics under compression are provided.

As expected, the pristine MWCNTS have some issues with agglomeration leading to track formation and little improvement of the composite properties, whereas silane  functionalised  CNTs  have  improvement of the strength and  allow to remain good  deformation characteristics.

It would be recommended in the future to provide modeling experiments at higher CNT contents and also to evaluate other   carbonaceous materials such as    graphene nanoplatelets. 

Author Response

No

Comments/ Recommendations/ Suggestions/ Queries

from the Reviewer

Feedback from the Authors

1.

This paper presents an interesting study of pristine and silane functionalised MWCNTs as reinforcing agents of hybrid fibre material composites. There is high practical importance of jute, kenaf and other textile material application studies. In the last decade, several studies on the application of MWCNTs and other carbonaceous materials have been reported, including composites based on kenaf and other polymers. It would be recommended to better explain the innovation of this study comparing with the other reported studies. While 3-aminopropyltriethoxysilane is well known coupling silane agent, this study presents interesting data of its application for the improvement of MWCNT dispersion in kenaf/glass fibre composite system. The composites have been analysed at wide ratios of pristine, acid treated and silane coupled   CNTS (0-1 wt.%) and all the necessary experimental conditions for the surface, structure analysis and evaluation of stress-strain characteristics under compression are provided. As expected, the pristine MWCNTS have some issues with agglomeration leading to track formation and little improvement of the composite properties, whereas silane functionalised CNTs have improvement of the strength and allow to remain good deformation characteristics. It would be recommended in the future to provide modeling experiments at higher CNT contents and also to evaluate other carbonaceous materials such as graphene nanoplatelets.

Thank you for the comments and recommendations given by the reviewer.

The innovations or novelty of this study when compared to the other reported studies in the literature are as follows:

1.     In this paper, unidirectional (UD) kenaf reinforced epoxy polymer composites were fabricated and the mechanical properties under compression loading were analysed. It is very limited study being conducted by other researchers on the unidirectional kenaf composites. Properties data of Unidirectional direction fibres composites are very important in order to get the maximum strength, elastic modulus and elastic strain of various types kenaf based composite system. These basic data are valuable in predicting the properties of off-axis layup composites and also reference data for all other composite systems.

2.     In this study, compression test was conducted in order to analysed the strength, elastic modulus and elastic strain. In literature, it is very limited study on compression properties of fibre reinforced polymer (FRP) composites. Tensile and flexural properties are among the properties that usually reported by numerous researchers in the literature. Compression properties are very important in designing FRP composites. The compressive strength of FRP composite is usually 60% lower than the tensile strength. Therefore, compressive strength need to be taken into account for designing FRP composite structures/components. The analysis of the fractures samples showed interesting fracture mechanisms, i.e. Fibre kinking and fibre shearing.

3.     In the fabrication of the FRP composites, this paper reports an advanced technique of dispersing MWCNT, i.e. using the high shear roll milling technique.

4.     This study contributes to new knowledge on optimum surface treated MWCNT loading/weight fraction in order to improve the FRP properties. The treatment methods of the MWCNT surface were described in details for other researchers use as reference.

All these innovations or important points have been added in revised manuscript

Reviewer 2 Report

The article studied the effect of chemically treated MWCNT on the mechanical properties of kenaf and kenaf/glass composites. It is an interesting research towards composites derived from sustainable material from natural source. The article is well-written with promising results. I suggest for publication with below minor changes,

  1. The ruler on TEM figures of Figure 3-5 is not readable. The number next to the ruler is too small.
  2. It seems that the TEM images of ACNT and SCNT also got some dark spots. Are they entanglement? Also, if that’s possible, an image with lower magnification but large field could help to display a “entanglement density”.
  3. It is a bit confusing when you mentioned about converting the MWCNT from hydrophobic to hydrophilic for better compatibility to resin, but page 2 line 55 a statement about the “hydrophobicity of resin”. Please comment to clarify.
  4. I did not see the volume fraction of fibers your composites in the experimental section. What is the volume you use for those samples? Are they the same for K, G and K/G composites? In another word, if you use 5-7% of glass fiber, is that a 1:1 replacement of kenaf? As the comparison on page 10 should be based on same or similar vol.%.
  5. Have you performed volume fraction analysis after the composite was made?

Author Response

No

Comments/ Recommendations/ Suggestions/ Queries

from the Reviewer

Feedback from the Authors

1.

The ruler on TEM figures of Figure 3-5 is not readable. The number next to the ruler is too small.

The ruler and scale number have been redrawing as the number next to the ruler is too small. For a better representation of the dispersion state, the TEM images have been changed/revised with lower magnification as advised/requested in question no 2. (Refer: pages 8-9 in the paper). The text related to TEM images has also been improved/revised (Refer: page 7 in the paper)

2.

It seems that the TEM images of ACNT and SCNT also got some dark spots. Are they entanglement? Also, if that’s possible, an image with lower magnification but large field could help to display a “entanglement density”.

As suggested by the reviewer, the higher magnification of 105000x TEM images in figure 3, 4 and 5 have been replaced with lower magnification of 43000x TEM images. Some dark spots are not representing the entangled cluster of CNTs. This dark spot appears because of the overlapping of the homogeneously dispersed CNTs. The TEM samples were prepared using ultra-microtome cutter. These samples have an average thickness of 85nm. The average diameter of the CNT is about 11nm therefore when the electron is transmitted through the 80nm thickness sample, it will cause dark spot on the image due to the overlapping/intersecting CNT in the epoxy. Therefore, the dark spots show overlapping MWCNTs that dispersed homogeneously in the epoxy. 

3.

It is a bit confusing when you mentioned about converting the MWCNT from hydrophobic to hydrophilic for better compatibility to resin, but page 2 line 55 a statement about the “hydrophobicity of resin”. Please comment to clarify.

For a better understanding, the sentence has been rephrased as follows:

”Chemical modification of MWCNT results in the presence of functional groups or epoxy ended-groups that are covalently bonded on the surface of the MWCNTs. The presence of these functional groups modifies the surface characteristics of the MWCNTs and improves the dispersion of MWCNTs within epoxy resin.” (Refer: Page 2)

4.

I did not see the volume fraction of fibers your composites in the experimental section. What is the volume you use for those samples? Are they the same for K, G and K/G composites? In another word, if you use 5-7% of glass fiber, is that a 1:1 replacement of kenaf?

The same kenaf fibre weight fraction was used in the fabrication of kenaf composites and hybrid composites. For the hybrid K/G composites, the woven glass fibres were added at the outside layer of the kenaf composites. The effect of adding glass at outer layers on the compressive properties of the kenaf composites was studied. Therefore, the presence of woven glass does not replace the kenaf.   

5.

Have you performed volume fraction analysis after the composite was made?

The volume fraction analysis, after composite was made, was performed via image analyser, bun off and acid digestion method.